

# The expression and clinical significance of GADD45A in breast cancer patients

Junnan Wang[1,*], Yiran Wang[2,*], Fei Long[3], Fengshang Yan[1], Ning Wang[3] and Yajie Wang[3]

[1] Basic Medical College, Navy Medical University, Shanghai, China
[2] Changhai Hospital, Navy Medical University, Shanghai, China
[3] Department of Oncology, Changhai Hospital, Navy Medical University, Shanghai, China
[*] These authors contributed equally to this work.

## ABSTRACT

**Background.** Growth arrest and DNA-damage-inducible protein 45 alpha (GADD45A) was previously found to be associated with risk of several kinds of human tumors. Here, we studied the expression and clinical significance of GADD45A in breast cancer.

**Methods.** We performed an immunohistochemical study of GADD45A protein from 419 breast cancer tissues and 116 adjacent non-neoplastic tissues.

**Results.** Significantly high GADD45A expression were observed in breast cancer tissues compared with adjacent non-neoplastic tissues ($P < 0.001$) and were independently correlative with estrogen receptor negative ($P = 0.028$) and high Ki-67 index ($P < 0.001$). Kaplan–Meier survival analysis revealed that patients with high GADD45A expression levels had a worse long-term prognosis in triple negative breast cancer ($P = 0.041$), but it was not an independent prognostic factor in multivariate analysis ($P = 0.058$).

**Conclusions.** GADD45A expression levels are significantly correlative with estrogen receptor status and Ki-67 index in human breast cancer. Patients with triple negative breast cancer might be stratified into high risk and low risk groups based on the GADD45A expression levels.

Corresponding author
Yajie Wang, yayachris@smmu.edu.cn, wangyjscholar@sina.com

## BACKGROUND

Breast cancer has become the most common malignant tumor threatening women's health whether in the developed countries or in developing countries. In the United States, there are an estimated 252,710 female cases diagnosed with breast cancer in 2017 (*Siegel, Miller & Jemal, 2017*). Meanwhile the incidence of breast cancer in China presents the trend of rising rapidly, over 3.9% annual percentage change from 2000 to 2011 (*Chen et al., 2016*). However, the pathogenesis of breast cancer, possibly related to disorder of cell cycle and genetic abnormalities, is not clear yet. Therefore, understanding the connection of biological characteristics and clinicopathological significance is an important aspect for investigating the pathogenesis and identifying therapeutic targets.

Growth arrest and DNA-damage-inducible protein 45 alpha (GADD45A) is a downstream target gene of p53 and BRCA1 (breast cancer susceptibility gene 1), which are

two of the most important genes maintaining the stability of the genome and suppressing the development of tumor through various mechanisms (*Li et al., 2017b*). As a member of the GADD45 family of genes that are known as stress sensors, GADD45A modulates the cellular response to a variety of stress conditions, including genotoxic and oncogenic stress (*Cretu et al., 2009*). Also, some research showed GADD45A can promote or suppress breast carcinoma development depending on different signaling pathways (*Carrier et al., 1999*; *Miki et al., 1994*).

Additionally, the correlation of GADD45A expression and clinicopathologic factors is not clear in breast cancer thus far. The aim of the current study is to investigate the significance of GADD45A in breast cancer, including the correlation with clinicopathologic factors and long-term survival situation.

## MATERIALS AND METHODS

### Tissue samples and clinical data

The study retrospectively collected 419 cases of tissue specimens from breast cancer patients, undergoing the operation from February 2005 to October 2007, at Changhai Hospital, Shanghai, China. Patients with other primary tumor sites were excluded. Clinicopathologic data was recorded from the patients' medical records including age at diagnosis, molecular classification (ST Gallen), TNM stage (AJCC), pathology types, histological grade (Elston-Ellis grade) and clinical outcomes. Cutoff for positivity was set at $\geq 1\%$ of cells staining positively for ER or PR. HER-2 status was assessed using the HercepTest$^{TM}$ according to the manufacturer's instructions. HER-2 positivity was defined as a 3+ score on IHC in >10% of invasive tumor cells. Equivocal IHC cases (2+ score or 3+ score in $\leq 10\%$ of invasive tumor cells) were submitted to FISH analysis. A ratio of HER-2 signals to chromosome 17 signals >2.0 was used as the cutoff to define HER-2 amplification. Ki-67 index greater than 20% was classified as high. In short, Luminal A was defined as ER+ or PR+, HER2- and low Ki67, Luminal B as ER+ or PR+ and HER2+ or high Ki67; HER2 positive as HER2+ and ER- and PR-; basal-like as HER2-/ER-/PR-. In addition, 116 cases of adjacent non-neoplastic tissue samples were also collected from mammoplasties as a control group.

GADD45A color reaction/positive staining were observed in 419 tumor tissues and 116 adjacent non-neoplastic tissues. The study was approved by the Ethics Committee of Biomedicine, Navy Medical University. The samples were used after obtaining informed written consents from all included patients.

### Tissue microarray and immunohistochemistry

The breast cancer tissue microarray (TMA) was obtained from the surgically resected tissues. Biopsies were fixed in 10% neutral formalin. TMA blocks were made by tissue arraying instrument (Beecher Instruments, Sun Prairie, WI, USA). Cylinders cores (1.5 mm in diameter) of tissue blocks were extracted from the center of the tumor, and then arranged into blank recipient paraffin blocks. The TMA blocks were cut into 4-µm sections and treated for immunohistochemistry (IHC). Following deparaffinization and rehydration of the tissues sections, antigen retrieval was performed in the pressure cooker in citrate buffer, pH 6.0, for 30 min at a sub-boiling temperature. Endogenous peroxidase was blocked with

3% peroxide for 10 min. Primary GADD45A antibody (1:200; catno. Sc-792; Santa Cruz Biotechnology, Dallas, TX, USA) were used on tissue sections for immunohistochemical staining. For a negative control, we replaced the primary antibody rabbit immunoglobulin G or normal murine with the same dilution.

The GADD45A staining results, independently reviewed and scored in triplicate by two pathologists, were assessed by the staining intensity and the percentage of positively stained cells of each tissue sample. The final GADD45A expression score can be calculated through multiplying the positive cells percentage score (0, <10%; 1, 10–25%; 2, 26–50%; 3, 51–75%; and 4, >75%) by the staining intensity score (0, negative; 1, weak; 2, moderate; 3, strong staining). The expression situation of tissue sections were evaluated by the total score (>five, high expression; ≤5, low expression). To calculate a mean score, total 10 independent fields with high magnification ($\times 200$) were observed.

### Statistical analyses

Statistical analyses were conducted using the IBM SPSS Statistics (for Windows, version 21.0). The Pearson $\chi 2$ test or Fisher's exact test was used to compare the GADD45A expressions between breast cancer tissues and adjacent non-neoplastic tissues and to confirm the correlations between clinicopathological factors. In the combined analyses, all the datasets were pooled and the odds ratios and $P$-values were estimated with logistic regression model. To assess the effect of GADD45A expression and other factors on overall survival (OS), different Kaplan–Meier curves were plotted with log-rank test. A multivariate Cox regression model was fitted to test the independent contribution of each variable to the patient's OS. For all data, differences were considered statistically significant when $P < 0.05$. Separate analysis was performed for the subtypes of the breast cancer.

## RESULTS

### Analysis of clinical data

The color reaction/positive staining of GADD45A was observed in 419 tumor tissue samples and 116 adjacent non-neoplastic tissue samples (Fig. 1). Among these 419 tumor tissue samples, 365 (87.1%) were invasive ductal carcinoma (IDC) and 54 (12.9%) were non-IDC. As for molecular types, there were 74 (17.7%) for luminal A, 206 (49.2%) for luminal B, 70 (16.7%) for Her-2 and 69 (16.5%) for triple-negtive (TN) subtype breast cancer. As for histological grades, there were 291 (69.5%) for grade I–II and 128 (30.5%) for III. In addition, there were 113 (27.0%) for TNM I, 233 (55.6%) for TNM II, and 73 (17.4%) for TNM III, respectively.

### Expression of GADD45A in breast carcinoma and normal tissues

The cytoplasmic staining for GADD45A was identified in the breast carcinoma tissues and adjacent non-neoplastic tissues. Overall, high expression of GADD45A was found in 72.3% (303/419) cancer tissues and 37.9% (44/116) adjacent noncancerous tissues. The percentage of high GADD45A expression of breast carcinoma tissues was obviously higher than that of adjacent noncancerous tissues and the difference was statistically significant ($P < 0.001$, Table 1).

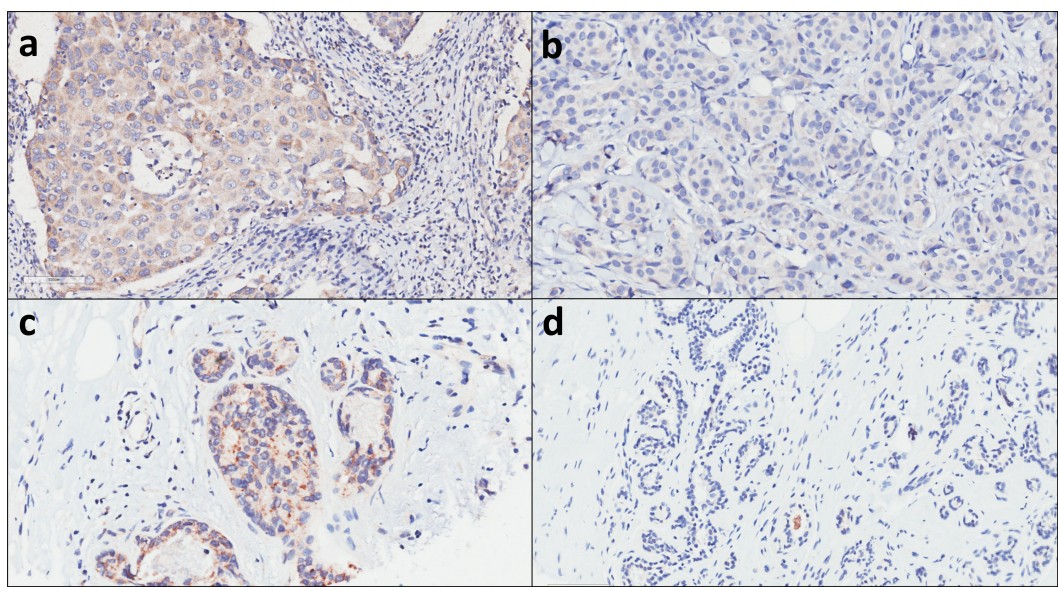

**Figure 1** **GADD45A expression levels are elevated in human breast cancer samples compared to adjacent non-neoplastic tissues.** Representative images from immunohistochemical analysis of GADD45A on (A) high expression in breast cancer tissues, (B) low expression in breast cancer tissues, (C) high expression in adjacent non-neoplastic tissues and (D) low expression in adjacent non-neoplastic tissues.

**Table 1** **Expression of GADD45A breast carcinoma tissues and adjacent non-neoplastic tissues.**

| Characteristics | GADD45A expression | | Total | $\chi2$ | $P$ |
|---|---|---|---|---|---|
| | High (%) | Low (%) | | | |
| Breast carcinoma tissues | 303(72.3) | 116(27.7) | 419 | | |
| Adjacent non-neoplastic tissues | 44(37.9) | 72(62.1) | 116 | 47.125 | <0.001 |

## Correlation between GADD45A expression and clinical pathological characteristics

The clinical pathological factors used in the current study include the following: age at diagnosis, TNM stage, pathology types, histological grade, molecular subtypes, ER (estrogen receptor), PR (progesterone receptor), HER2 (human epidermal growth factor receptor-2) and Ki-67 index. We found a significant correlation between GADD45A expression and ER status, Ki-67 index, histological grade and molecular subtypes but not between GADD45A expression and age at TNM stage, pathology types, PR and HER2. The high expression rate of GADD45A was 78.0% (131/168) in ER negative group which is 68.5% (172/251) in ER positive group and the difference was statistically significant ($P = 0.034$, Table 2). The high GADD45A expression rate of Ki-67 index high-level group ($\geq$20%) (83.1%, 207/249) was apparently higher than that of Ki-67 index low-level group (<20%) (56.2%, 96/170), ($P < 0.001$). Meanwhile, a positive correlation was found between GADD45A expression and histological grade ($P = 0.013$). It was also found between GADD45A expression and molecular subtypes ($P < 0.001$).

**Table 2 Correlation between high/low GADD45A expression and clinic pathological factors in 419 breast cancer tissues.**

| Characteristics | GADD45A expression | | Total | χ2 | P |
|---|---|---|---|---|---|
| | High (%) | Low (%) | | | |
| **Age at diagnosis (years)** | | | | | |
| ≤50 | 119 (70.0) | 51 (30.0) | 170 | | |
| >50 | 184 (73.9) | 65 (26.1) | 249 | 0.766 | 0.382 |
| **TNM stage** | | | | | |
| I | 77 (68.1) | 36 (31.9) | 113 | | |
| II | 168 (72.1) | 65 (27.9) | 233 | | |
| III | 58 (79.5) | 15 (20.5) | 73 | 2.846 | 0.241 |
| **Pathology type** | | | | | |
| IDC | 269 (73.7) | 96 (26.3) | 365 | | |
| non-IDC | 34 (63.0) | 20 (37.0) | 54 | 2.708 | 0.100 |
| **Histological grade** | | | | | |
| I–II | 200 (68.7) | 91 (31.3) | 291 | | |
| III | 103 (80.5) | 25 (19.5) | 128 | 6.120 | 0.013 |
| **Molecular subtype** | | | | | |
| Luminal A | 40 (54.1) | 34 (45.9) | 74 | | |
| Luminal B | 157 (76.2) | 49 (23.8) | 206 | | |
| HER2-enriched | 56 (80.0) | 14 (20.0) | 70 | | |
| Triple negative | 50 (72.5) | 19 (27.5) | 69 | 23.584 | <0.001 |
| **ER** | | | | | |
| positive | 172 (68.5) | 79 (31.5) | 251 | | |
| negative | 131 (78.0) | 37 (22.0) | 168 | 4.489 | 0.034 |
| **PR** | | | | | |
| positive | 163 (70.3) | 69 (29.7) | 232 | | |
| negative | 140 (74.9) | 47 (25.1) | 187 | 1.098 | 0.295 |
| **HER2** | | | | | |
| positive | 134 (75.7) | 43 (24.3) | 177 | | |
| negative | 169 (69.8) | 73 (30.2) | 242 | 1.760 | 0.185 |
| **Ki-67** | | | | | |
| ≥20% | 207 (83.1) | 42 (16.7) | 249 | | |
| <20% | 96 (56.2) | 74 (43.5) | 170 | 35.871 | <0.001 |

**Notes.**

IDC, invasive ductal carcinoma; ER, estrogen receptor; PR, progesterone receptor; HER2, human epidermal growth factor receptor 2.

ER status, Ki-67 index and histological grade were selected into logistic regression model analysis (Molecular subtype was not included because its effect was similar with ER status). The results from multivariate analysis confirmed that ER negative and Ki-67 ≥20% were closely related to high GADD45A expression (Table 3).

## Survival analysis

Kaplan–Meier curves and log-rank tests were used to perform survival analysis to identify the correlation between survival and GADD45A expression. The last follow-up was carried out in July 2017. Table 4 showed GADD45A expression was not related to survival when all

**Table 3  Multivariate analyses by logistic regression analysis.**

| Dependent variable | Independent variable | SE | *P* value | OR | 95 % CI |
|---|---|---|---|---|---|
| GADD45A expression | ER negative | 0.241 | 0.028 | 1.697 | 1.058–2.723 |
| | Ki-67 ≥20% | 0.231 | <0.001 | 3.861 | 2.455–6.074 |

**Table 4  Univariate analyses of OS and GADD45A expression of different types patients.**

| Analyses for GADD45A | $\chi2$ | *P* value |
|---|---|---|
| All types as total | 0.062 | 0.803 |
| Luminal A | 2.011 | 0.156 |
| Luminal B | 3.536 | 0.060 |
| HER2 | 1.565 | 0.211 |
| TN | 4.157 | 0.041 |

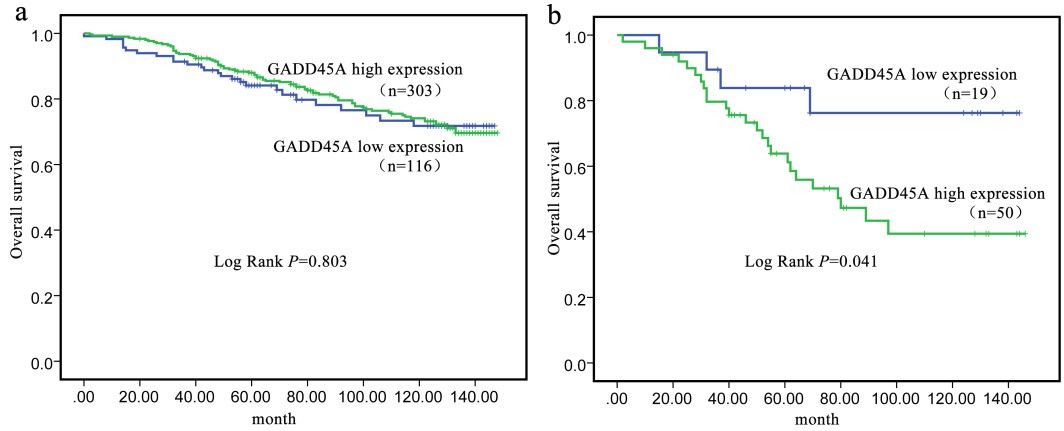

**Figure 2  Kaplan–Meier curve of overall survival in relation to GADD45A protein expression.** (A) breast cancer; (B) TN subtype breast cancer.

types as total ($P = 0.803$). However, in a survival analysis for TN subtype patients, the low GADD45A expression group had a mean survival period of 119.61 months (SE: 10.81, 95% CI [98.43–140.79]). However, the high GADD45A expression group had a mean survival period of 88.73 months (SE: 7.78, 95% CI [73.48–103.97]) ($P = 0.041$). Moreover, the 5-year and 10-year survival rate were 94.7% and 80.5% for the high GADD45A expression group while 79.7% and 68.5% for the low GADD45A expression group, respectively. Kaplan–Meier survival curves were performed in Fig. 2.

In the univariate analysis, the TNM stage and GADD45A were found to be significant independent predictors of OS, respectively (Table 5). However, in the multivariate analysis, only the TNM stage was identified the significant independent predictor of OS in the TN subtype patients (Table 5).

**Table 5  Univariate analyses and multivariate analyses overall survival for TN breast cancer patients.**

| Characteristics | Univariate analyses | Multivariate analyses | | |
| --- | --- | --- | --- | --- |
| | *P* value | HR | 95% CI | *P* value |
| Age (>50 vs. ≤50) | 0.158 | NA | NA | NA |
| TNM stage (III vs I/II) | 0.001 | 3.720 | 1.637–8.452 | 0.002 |
| Pathology type (non-IDC vs IDC) | 0.445 | NA | NA | N |
| Histological grade (III vs I/II) | 0.251 | NA | NA | NA |
| Ki-67 (>20% vs. ≤20%) | 0.060 | NA | NA | NA |
| GADD45A (Positive vs. Negative) | 0.041 | 2.792 | 0.967–8.060 | 0.058 |

# DISCUSSION

As GADD45A plays a role in genomic stability and tumorigenesis, some studies explored the role of GADD45A in various cancers, such as prostate cancer (*Reis et al., 2015*), ovarian cancer (*Yuan et al., 2015*), esophageal cancer (*Ishiguro et al., 2016*) and malignant gliomas (*Cui et al., 2017*), which revealed different GADD45A expressions in various carcinoma tissues. In research on esophageal squamous cell carcinoma (ESCC) (*Wang et al., 2012*) and another article on breast cancer (*Tront et al., 2013*), GADD45A protein levels were increased significantly in tumor tissues than those in normal tissues. In addition, a recent study showed that the expression of GADD45A was upregulated in human non-small-cell lung cancer (NSCLC) cells through a microRNA-138 related post-transcriptional pathway (*Li et al., 2017a*).

Compared with the previous study for GADD45A in breast cancer (*Tront et al., 2013*), our study had a larger sample analysis for the correlations between clinicopathological factors and GADD45A expression. We also showed the correlation with PR and ER, respectively, while Tront's study only analyzed tissues of ER+/PR+. Moreover, our study was the first study estimating the prognostic significance of GADD45A in triple negative breast cancer. We found that GADD45A expression was upregulated in breast cancer tissues. DNA damage was a major source of potentially mutagenic lesions leading carcinogenesis (*Gao, Herman & Guo, 2016*). GADD45A was shown to participate actively in DNA repair mechanisms (*Wingert & Rieger, 2016*). Therefore, its upregulation may be due to extensive DNA repair in breast cancer tissues. Beyond that, a study showed that GADD45A promotes Myc-driven breast carcinogenesis by negatively regulating matrix metalloproteinase 10 (MMP10) through GSK-3β (Glycogen synthase kinase 3 beta)/β-catenin signaling, resulting in increased tumor vascularization and growth (*Tront et al., 2010*). In our opinions, this mechanism may result for the high GADD45A expression in breast cancer tissues.

In our study, ER negative and high Ki-67 index were demonstrated the significant association with GADD45A expression. ER-negative breast cancer, which had less treatment options such estrogen-blocking endocrine therapy, had more malignancy behaviors than ER-positive tumors. Ki-67 index was a cellular marker strictly associated with cell proliferation and emerged as a key discriminative feature of malignant phenotype in breast cancer (*Pathmanathan et al., 2014*). The above factors all suggested high malignant degree.

From this, we considered that GADD45A may be a biomarker for the malignancy grade of breast cancer tissue. Our results were opposite to the results presented by *Tront et al. (2013)* which showed that high GADD45A levels were associated with luminal A and luminal B subtypes but not in triple negative tissues. The discrepancy could be generated by the higher number of tissues analyzed in our study compared to the previous one (419 vs 45). In addition, we explored the correlation with ER and PR status separately while (*Tront et al., 2013*) analyzed only tissue where they were both positive, as mentioned above. As a result of these advantages, our study was a more credible and detailed exploration for the correlations between clinicopathological factors and GADD45A expression compared previous study.

For further evaluation, we assessed the effect of GADD45A expression on OS. We demonstrated that patients with higher GADD45A expression levels had a poor long-term prognosis for TN subtype based on Kaplan–Meier survival curves. However, GADD45A expression was not an independent prognosis factor in TN type breast cancer in multivariate analysis. The finding may suggest that GADD45A expression affect the prognosis through the effects of other factors but not the independent action, such as the effect of Ki-67 index, which was reported as an independent variable associated with prognosis in TN subtype patients (*Adamo et al., 2017*; *Pan et al., 2017*; *Ricciardi et al., 2015*). In addition, we found that TNM stage independently influenced the prognosis of TN breast cancer patients. The similar conclusion was determined in previous studies (*Han et al., 2017*; *Yu et al., 2013*; *Yu et al., 2014*).

The limitations of current study included the deficiency of follow-up in chemotherapy regimens of patients. It was difficult to analyze the influence of chemotherapy regimen on prognosis of breast cancer. The relationship between different chemotherapy treatments and prognosis in breast cancer should be the focus of the future study and further analyses in biological mechanisms about tumor progression are need. Moreover, the possible selective bias due to the small sample size and inclusion of only postoperative patients may exist in TN breast cancer group. Despite the limitations, the correlation between GADD45A expression and certain clinicopathological factors in breast cancer can be verified.

## CONCLUSIONS

In present study, we explored the expression and clinical significance of GADD45A in breast carcinoma. High GADD45A was observed in breast cancer tissues compared with adjacent non-neoplastic tissues. ER negative and high Ki-67 index were correlative with high GADD45A expression. Patients with higher GADD45A expression levels had a poorer long-term prognosis in TN type breast cancer. However, only the TNM stage was the independent prognosis factor in TN type breast cancer. Our study suggested that GADD45A may play a role in breast cancer pathogenesis and may eventually help in understanding the biological mechanisms affecting tumor progression.

## INSTITUTIONAL ABBREVIATIONS

| | |
|---|---|
| GADD45A | DNA-damage-inducible protein 45 alpha |
| BRCA1 | Breast cancer susceptibility gene 1 |
| TMA | Tissue microarray |
| OS | Overall survival |
| IDC | Invasive ductal carcinoma |
| TN | Triple-negtive |
| ER | Estrogen receptor |
| PR | Progesterone receptor |
| HER2 | Human epidermal growth factor receptor-2 |
| ESCC | Esophageal squamous cell carcinoma |
| NSCLC | Non-small-cell lung cancer |
| MMP10 | Matrix metalloproteinase 10 |
| GSK-3$\beta$ | Glycogen synthase kinase 3 beta |

### Funding

Funding for this study was provided by the National Natural Science Foundation of China [grant number 81572591]. The funders had no role in study design, data collection and analysis, decision to publish, or preparation of the manuscript.

### Grant Disclosures

The following grant information was disclosed by the authors:
National Natural Science Foundation of China: 81572591.

### Competing Interests

The authors declare there are no competing interests.

### Author Contributions

- Junnan Wang and Yiran Wang analyzed the data, prepared figures and/or tables, authored or reviewed drafts of the paper, approved the final draft.
- Fei Long and Fengshang Yan performed the experiments, authored or reviewed drafts of the paper, approved the final draft.
- Ning Wang performed the experiments, contributed reagents/materials/analysis tools, prepared figures and/or tables, authored or reviewed drafts of the paper, approved the final draft.
- Yajie Wang conceived and designed the experiments, authored or reviewed drafts of the paper, approved the final draft.

### Human Ethics

The following information was supplied relating to ethical approvals (i.e., approving body and any reference numbers):

The Ethics Committee of Biomedicine, Changhai Hospital, Second (Navy) Military Medical University granted study approval.
## Data Availability

The raw data are provided in a Supplemental File.

## Supplemental Information

Supplemental information for this article can be found online at http://dx.doi.org/10.7717/peerj.5344#supplemental-information.

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
