# Peer review of "The expression and clinical significance of GADD45A in breast cancer patients"

_PeerJ, doi:10.7717/peerj.5344_

## Round 0.1 · original submission · Major Revisions

The manuscript should be deeply revised according to the referees' comments (especially referee 1). I strongly recommend to improve the English language used.

Reviewer 1 ·

Basic reporting

The manuscript “The expression and clinical significance of GADD45A in triple negative breast cancer” by Wang et al. presents the expression of GADD45A in breast cancer as potential new prognostic biomarker in particular in triple negative subtype. Breast Cancer remains the most common malignancy and the major cause of cancer death among women worldwide and new achievements in prevention and treatment, including reliable new prognostic and predictive biomarkers, are still required. This study is therefore well focused on the current clinical needs.

However some issues had to be addressed for publication.

The first concerns regards the English language used, that should be improved to ensure a can clear understanding of the text and the conclusions. Some paragraphs (such as lines 172-181) are confused and re-phrasing could be useful to clearly explain and correlate the results with previous findings. I suggest a complete review of manuscript aimed to better organize the sentences and support the conclusions. Some editings are needed as well (grammar (ex line 198) or table (Ki67 value in Table 4).

The literature listed is sufficient to describe the state of the art and define the study goals, but a deeper discussion of the current results in comparison with previous finding is needed. In particular, authors seems to claim that this is the first study where clinicopathologic features are correlated with GADD45A expression. However, as they correctly reported, Tront et al. in 2013 performed similar investigations. For example both the studies demonstrated that GADD45A expression is higher in tumor compared to normal/adjacent tissue. I suggest to insert some sentences to put in evidence the differences between the two studies and to underline the novelty of their own.
The paper results are consistent with the study goals.
Raw data are provided as for PeerJ policy requirements (even only 1 TMA is shown); ethical documentation presented only in Chinese.

Experimental design

The study aims to correlate GADD45A expression obtained by IHC analysis to clinical pathological feature. This a basic research that potentially could add new information for the understanding of breast cancer biology. It has the great advantage to present a large cases collection, well divided in subtypes, with a long patients’ follow up.
Material and Methods:
Correct parameters and references are reported for breast cancer characterization (the definition/evaluation of ER/PgR and Her 2 expression/positivity are the currently used in clinic).
However I have some concerns about the Ki67 cut off. Authors classify Ki67 index> 14% as high. The accepted cut off value for ki67 is 20%, as from St Gallen consensus (Personalizing the treatment of women with early breast cancer: highlights of the St Gallen International Expert Consensus on the Primary Therapy of Early Breast Cancer 2013. Goldhirsch A, Winer EP, Coates AS, Gelber RD, Piccart-Gebhart M, Thürlimann B, Senn HJ; Panel members. Ann Oncol. 2013 Sep;24(9):2206-23.)
Therefore this value should be used to discriminate between high and low Ki67 expression. Could the authors explain the reason why they choose 14% as Ki67 cut off? This value is quite important since from their analysys this parameter correlates with GADD45A expression. On note, strangely almost half of triple negative cases analysed in this work (30 vs a total of 79 tissues) present ki67 values lower that 14%, while, as the authors remarked in the discussion, ki67 is normally high in triple negative breast cancer.
Regarding ICH analysis, material and methods section is quite vague. A very poor description of the procedure is presented (line 839 “standard protocol” should be better described).
Moreover normally sections are analysed in triplicate by at least two independent pathologists. It’s not evident if the authors follow this correct laboratory practice to analyse the tissues.

Validity of the findings

The authors conclude that ER negative, high histological grade and high Ki-67 index correlate with high GADD45A expression. These conclusions are opposite to the ones presented by Tront et al (2013) who showed that high GADD45A levels are associated with luminal A and luminal B subtypes and that in triple negative tissue this expression is low. In the same way Tront and coworkers didn’t find any correlation with the histological grade, that here resulted significant.
Could the authors provide an explanation for these differences?
This discrepancy could be generated by the higher number of tissues analyzed in this work compared to the previous one (419 vs 45). Wang et al were also able to shown the correlation with PgR and ER separately while Tront et al analysed only tissue where they were both positive. This point could be taken into account to discuss the results. Comments on these data could be inserted in the manuscript.
To add novel information to the field I suggest these majors
- Provide correlation with breast cancer subtypes. The information about GADD45A expression in regards with the different subtypes could be important and useful for clinicians.
- Repeat the statistical analysis considering ki67 cut off > 20% to confirm the data.
- Include a multivariate analysis. The author stated that GADD45A is an independent prognostic value for triple negative breast cancer subtypes. However, as they put in evidence in the discussion (line 184), this could be an effect of ki67 value. The author performed only univariate Pearson χ2 test or Fisher´s exact test to confirm the correlations between clinicopathological factors and GADD45A expression. A multivariate analysis is recommended to strengthen and confirm that GADD45A could be considered a prognostic marker for triple negative breast cancer.

Reviewer 2 ·

Basic reporting

No comment

Experimental design

I suggest to report if the series are consecutive

Validity of the findings

No comment

Additional comments

The paper is interesting but I suggest to mitigate the emphasis on the prognostic role in TNBC because of the small sample size

Reviewer 3 ·

Basic reporting

no comment

Experimental design

no comment

Validity of the findings

no comment

Additional comments

This manuscript identified GADD45A as a potential treatment target by performing immunohistochemistry in 419 breast cancer tissues and 116 adjacent non-neoplastic tissues. Although this manuscript has assessed a number of patient samples, this review has concerns of their less developed experiments and less reasonable conclusions.
GADD45 is known as a tumor suppressor. Although it is induced in breast cancer, it does not conclude that GADD45A is a druggable target of cancer treatment. The function of GADD45A should be addressed with further experiments. In addition, it is also important to check the mutations in breast cancer. If GADD45A is frequently mutated or GADD45A has new oncogenic functions in breast cancer especially TNBC, the assumption of the authors may be possible.
The authors refer the publication of GADD45A-promoted MYC-driven cancer to explain their result and hypothesis. However, GADD45A is more well-known as a negatively regulated MYC downstream target.
Furthermore, the clinical significance of GADD45A in TNBC must be intensively validated to overcome all allegations of GADD45A role in breast cancer.
Overall, this review concludes that this manuscript is not sufficiently developed and described to be published in PeerJ.

---

## Round 0.2 · accepted · Accept

Although the referee has reported some minor concerns, I think the manuscript can be accepted for publication.

Reviewer 1 ·

Basic reporting

Wang and co-workers evaluated the expression of GADD45A in breast cancer using IHC analysis.

One of the main concerns regarding the manuscript was the English language used, since some paragraphs were poorly written and confused. The authors deeply revised the text improving the language, making more understandable data description and relative conclusions. Some corrections are still necessary (ex see lines 145 and 225) but the overall form is acceptable

Experimental design

The study aims to correlate GADD45A expression obtained by IHC analysis to clinical pathological features. Compared to other studies it has the great advantage to present a large cases collection, patients’ follow up and survival information.

Material and Methods:
As better description of the IHC procedure (protocol and slides analysis) is now presented.
More importantly, the correlation between GADD45A and Ki67 was repeated considering the accepted cutoff of 20%, currently considered the reference value for this parameter. The revised analysis confirmed the previous findings.

Validity of the findings

In the first version of the manuscript the authors concluded that GADD45A was an independent prognostic factor for triple negative breast cancer, but this conclusion was not supported by a complete data analysis. As requested, statistical evaluation was repeated, including correlation with breast cancer subtypes and above all a multivariate analysis, that not confirmed the previous conclusion, therefore not providing the validation for GADD45A as independent biomarker for TNBC. Taking into account these results, the authors mitigated the emphasis on the clinical significance of GADD45A in TNBC. However, from this point of view the conclusion stated in the abstract regarding the possibility to use GADD45A expression level to stratify TNBC patients in risk categories, is probably still too speculative. In any case, all the requested revisions have been performed. Overall, a better description of the data and a more detailed and critical discussion of the results in respect of the state of the art was inserted, highlighting the different findings presented compared with similar studies, where opposite conclusion were shown. Considering a large patients' cohort analysed and the availability follow up data, the fact that authors were able to correlate GADD45A expression level with ER and PR status separately, this work could provide additional information that other investigators could take in to account. The editor could decide if the findings are interesting enough for publication.